# Superconductivity in 122-Type Pnictides without Iron

**Pan Zhang [1],\* and Hui-fei Zhai [2]**

1   Department of Physics, Ningbo University, Ningbo 315211, China
2   Department of Physics, Northwest University, Xi'an 710000, China; phyzhf@nwu.edu.cn
\*   Correspondence: zhangpanphy@zju.edu.cn; Tel.: +86-571-13735894256

**Abstract:** The exploration of superconducting iron-free pnictides with a $ThCr_2Si_2$-type or related structure and the study of their superconducting properties are important in order to get a deeper understanding of the pairing mechanism of 122 iron pnictides. Here we review the properties of 122-type iron-free pnictides superconductors with structures similar to that of $BaFe_2As_2$. Evidence of fully gapped nature of superconducting state has come from the specific heat and thermal conductivity measurements for $BaNi_2As_2$ and $SrNi_2P_2$, and nuclear magnetic and quadrupole resonance measurements for $CaPd_2As_2$ etc. Combined with the fact that no magnetism is observed in 122-type iron-free pnictides superconductors, the majority of evidence suggests that most of these compounds are conventional electron–phonon-mediated superconductors.

**Keywords:** 122-type; iron-free; superconductivity; charge-density wave; structural phase transition

## 1. Introduction

The discovery of superconductivity (SC) in LaFeAs(O,F) rekindled people's interest in iron and other transition metal pnictides. Among them, 122 family (e.g., $(Ba,K)Fe_2As_2$ [1]), which crystallize in the $ThCr_2Si_2$-type structure, has been intensely studied due to their wide range of dopings either by holes or electrons, and even by isovalent elements (for example, partial substitution of K for Ba, Co for Fe, or P for As can turn $BaFe_2As_2$ into superconductors [1–3]), as well as the easy growth of single crystals with sufficient sizes for various experiments. Except for chemical substitution, SC can also be achieved by applying external pressures in most of the undoped iron pnictides [4,5]. If Ba is replaced by magnetic rare-earth metals, this crystalline structure with alternating stacked layers along the *c*-axis will constitute a suitable platform for understanding the interplay between Kondo interaction and Ruderman–Kittel–Kasuya–Yosida (RKKY) interaction.

On the other hand, SC is also observed in other 122-type compounds formulated with $AT_2Pn_2$ ($A$ = alkaline-earth or rare-earth; $T$ = transition metals; $Pn$ = P, As or Sb) without iron [6–25]. It has been reported that these 122-compounds show a variety of interesting phenomena, such as SC [1], heavy fermion [26], densed Kondo behavior [27], coexistence of SC and anti-ferromagnetism (AFM)/ferromagnetism (FM)/spin-density-wave (SDW) or charge density wave (CDW) [1,13,28]. Moreover, studies on iron-based pnictides indicate that Fe has no magnetic moment, making the role of transition metals in $AT_2Pn_2$ fascinating. Although numerous studies have been conducted on the 122-type iron-based pnictides, to get further insight into the underlying physics of SC in $AT_2Pn_2$ system, including the role of the transition metal elements and other novel properties, many efforts on compounds with various transition metal elements have also been made, from both experimental and theoretical aspects. Here we attempt to give a simple introduction to iron-free $AT_2Pn_2$ superconductors and review the current status of this field. In the following sections, we review the materials and experiments for these 122-compounds which exhibit SC at low temperatures.

## 2. Results

### 2.1. Crystal Structure

The 122-type iron-free pnictides superconductors (hereafter abbreviated as 122-IFPSs) have a number of different crystal structures, while they share a similar layered structure with $BaFe_2As_2$ [1]. $ThCr_2Si_2$-type structure has been found in $SrNi_2As_2$, $LaRu_2P_2$, and $SrPd_2As_2$, etc. [6,7,10,11,15,17,23,24,29], whereas $SrPt_2As_2$, $BaPt_2As_2$, and $LaPd_2Sb_2$, etc. crystallize in $CaBe_2Ge_2$-type [13,16,21], a derivative of $ThCr_2Si_2$-type, structure at high temperature. Compared to the two-dimensional (2D) nature of the layer in $ThCr_2Si_2$-type structure, $CaBe_2Ge_2$-type structure is more 3D. For $SrPt_2Sb_2$ and $BaPt_2Sb_2$, their crystal structures are orthorhombic and monoclinic variants of $CaBe_2Ge_2$-type structure, respectively [18,19]. Structural phase transition is found in $BaNi_2As_2$ from a high-temperature tetragonal phase to a low-temperature triclinic phase at $T = 130$ K [30]. $APt_2Pn_2$-based ($A$ = Sr, Ba; $Pn$ = As, Sb) superconductors also exhibit a structural phase transition from tetragonal to a modulated orthorhombic variant of the $CaBe_2Ge_2$-type structure when lowering temperatures [13,16,18]. An exception is $BaPt_2Sb_2$, which shows a monoclinic structure below $T_s$ [19]. We show four representative crystal structures possessed by 122-IFPSs in Figure 1.

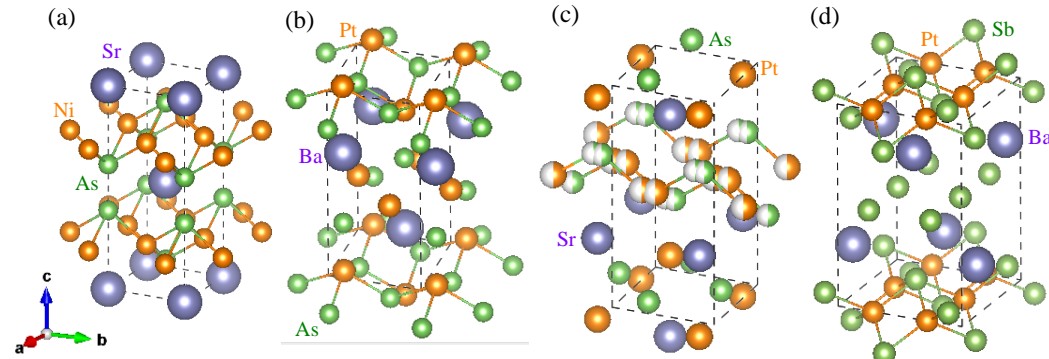

**Figure 1.** Four different kinds of crystalline structures that are usually found in 122-type iron-free pnictides (122-IFPSs). (**a**) $ThCr_2Si_2$-type; (**b**) $CaBe_2Ge_2$-type; (**c**) An orthorhombic variant of $CaBe_2Ge_2$-type, with space group P*mmn*; (**d**) A monoclinic variant of $CaBe_2Ge_2$-type, with space group C2/*m*.

### 2.2. Normal State Properties

The normal state resistivity of most 122-IFPSs exhibits metallic behavior without dramatic slope change that was observed in the parent compounds of iron-based superconductors due to spin-density wave (SDW). However, in Ni-based $BaNi_2As_2$ and $SrNi_2P_2$, an anomaly originated from structural transition was found in both specific heat and electrical resistivity measurements [8,10,30]. For $BaNi_2As_2$, clear anomalies can be observed in the temperature-dependent resistivity and specific heat curves at about 130 K [10]. Unlike the magnetic nature as a result of SDW instability (being driven by Fermi surface (FS) nesting) in $BaFe_2As_2$, the first-order transition in $BaNi_2As_2$ is probably caused by small reduction of conducting carriers due to the removal of several small FS sheets contributed dominantly from the As-As bonding and Ni-As antibonding below the structural transition ($T_s$) [31–33]. The angle-resolved photoemission spectroscopy (ARPES) study of the electronic structure further supports the idea that no SDW-type of magnetic ordering exists in $BaNi_2As_2$, as band folding cannot be found [34]. Furthermore, band shift shows a hysteresis similar to that observed in the resistivity data, which can be attributed to the lattice distortion below $T_s$. In comparison, the structural phase transition in $APt_2Pn_2$ ($A$ = Sr, Ba; $Pn$ = As, Sb) superconductors is identified to be related to the formation of a charge-density-wave (CDW) state [13,16,18,19,35]. Thus, these compounds were argued to exhibit the coexistence of SC and CDW instability.

SrPt$_2$As$_2$ experiences a structural phase transition at about 470 K [35,36]. A superstructure with a modulation wave vector **q** = 0.62**a**\* is formed below this temperature. Moreover, the modulation appears in the layers of the PtAs$_4$ tetrahedra, with the other atoms nearly unaffected. Figure 2a,b display the electron diffraction patterns taken along [001] zone axis of SrPt$_2$As$_2$ above (470 K) and below (300 K) the structural phase transition [37]. It is interesting to note that a series of satellite spots appear, aligned with the main diffraction spots at room temperature, while they vanish at 470 K. Those satellite spots can be characterized by the modulation wave vector **q** mentioned previously. The structural phase transition can also be seen in the resistivity curve, with the resistivity jump shown in Figure 2c. Furthermore, optical spectroscopy measurements conducted on SrPt$_2$As$_2$ reveal a removal of about 17% spectral weight at low frequency resulting from the structural phase transition [37], which suggests a reconstruction of FS. This picture also supports the formation of CDW state below the transition.

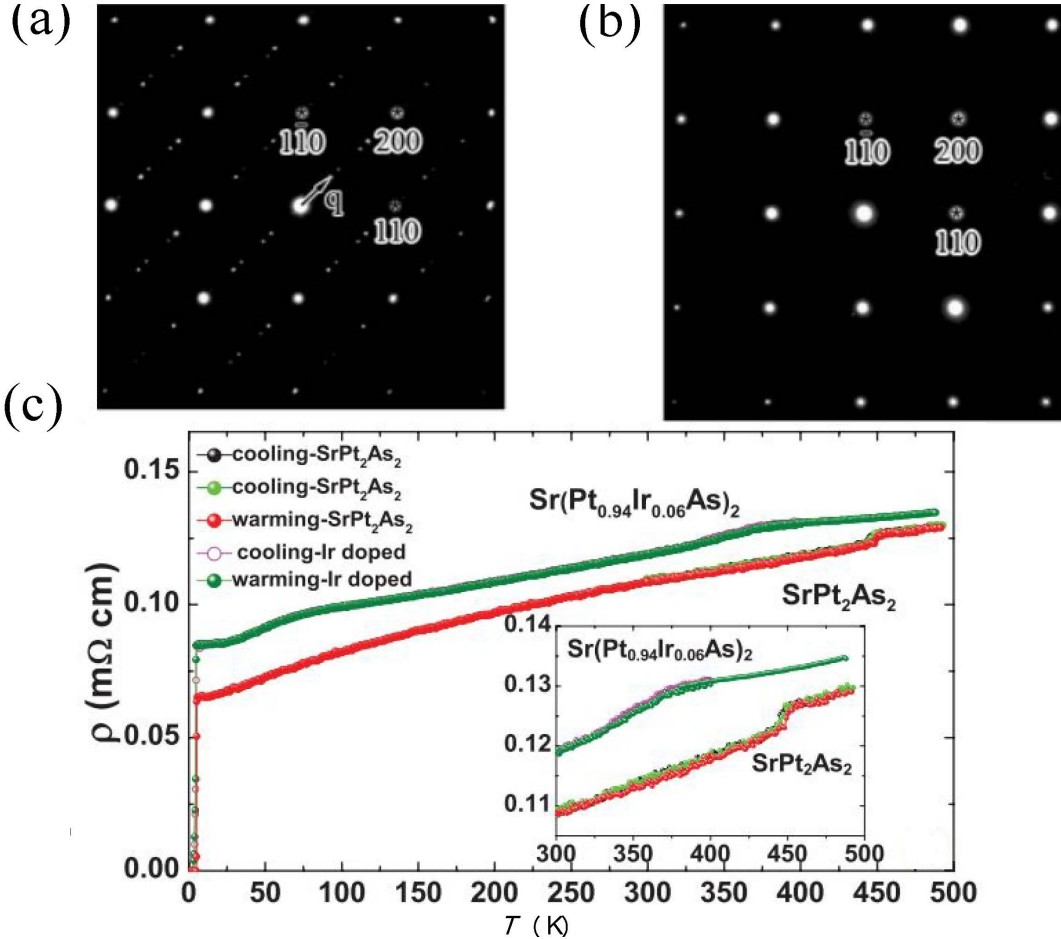

**Figure 2.** Electron diffraction patterns of SrPt$_2$As$_2$ single crystal performed at (**a**) 300 K and (**b**) 470 K. (**c**) The resistivity jump due to structural phase transition in SrPt$_2$As$_2$. Pictures are from Reference [37].

Similar structural phase transition which is likely to be associated with CDW instability has also been observed in BaPt$_2$As$_2$ and SrPt$_2$Sb$_2$ [16,18]. The first-order transition has been reported to show anomalies in both resistivity and specific heat curves at about 275 K in BaPt$_2$As$_2$ [16]. SrPt$_2$Sb$_2$ undergoes a structural phase transition between 134 K and 295 K, showing an anomaly in the resistivity curve with thermal hysteresis [18].

## 2.3. Superconducting Properties

Conventional Bardeen–Cooper–Schrieffer (BCS) SC is proposed for most 122-IFPSs [38–42]. We summarize the basic properties of 122-IFPSs in Table 1, including the $T_c$ for each compound. Due to the apparent lack of magnetism, 122-IFPSs could provide perfect systems to study the interplay between structure and SC. In addition, proximity to a structural phase transition is not a prerequisite for SC for these compounds, as evidenced by the absence of a structural phase transition for some of them [6,7,9,11,12,14,15,17,20–25].

**Table 1.** Properties of 122-IFPSs.

| Compound | $SrNi_2As_2$ | $BaNi_2As_2$ | $SrNi_2P_2$ | $BaNi_2P_2$ | $CaPd_2As_2$ | $SrPd_2As_2$ | $BaPd_2As_2$ | $LaPd_2As_2$ |
|---|---|---|---|---|---|---|---|---|
| space group | I4/mmm | I4/mmm | I4/mmm | I4/mmm | I4/mmm | I4/mmm | I4/mmm | I4/mmm |
| $a$ (Å) | 4.1374 (8) | 4.112 (4) | 3.951 | 3.947 | 4.2824 (2) | 4.3759 (1) | 4.489 (2) | 4.3027 (27) |
| $b$ (Å) | 4.1374 (8) | 4.112 (4) | 3.951 | 3.947 | 4.2824 (2) | 4.3759 (1) | 4.489 (2) | 4.3027 (27) |
| $c$ (Å) | 10.188 (4) | 11.54 (4) | 10.432 | 11.853 (2) | 10.0880 (4) | 10.1671 (3) | 10.322 (2) | 10.268 (14) |
| $T_S$ (K) | - | 130 | 325 | - | - | - | - | - |
| $T_c$ (K) | 0.62 | 0.68 | 1.4 | 2.7 | 1.27 (3) | 0.92 (5) | 3.85 | ~1 |
| $H_{c2}$ (T) | ~0.021 [c] | 0.09 [c] | 0.03[c] | 0.065 [c] | 0.157 | 0.073 | ~0.21 | 0.402 |
| | ~0.015 [ab] | 0.19 [ab] | - | 0.16 [ab] | - | - | - | - |
| $\gamma_0$ (mJ/(mol·K$^2$)) | 8.7 | 6.15 | 7.5 | 14 | 6.52 (2) | 6.43 (3) | 4.79 (2) | 5.56 |
| $\Delta C/\gamma T_c$ | ≃1.0 | 1.31 | 1.27 | 1.1 | 1.14 (3) | 0.77 (5) | - | 1.17 |
| References | [7] | [10,30,43] | [8,44] | [9,38,45,46] | [17] | [17] | [17,24] | [20] |

| Compound | $SrIr_2As_2$ | $BaIr_2As_2$ | $SrPt_2As_2$ | $BaPt_2As_2$* | $SrPt_2Sb_2$ | $BaPt_2Sb_2$ | $BaRh_2P_2$ | $BaIr_2P_2$ |
|---|---|---|---|---|---|---|---|---|
| space group | I4/mmm | I4/mmm | Pmmn | P4/nmm | P4/nmm | C2/m | I4/mmm | I4/mmm |
| $a$ (Å) | 4.068 (1) | 4.052 (9) | 4.46 | 4.564 | 4.603 | 6.70156 (10) | 3.9308 (3) | 3.9469 (8) |
| $b$ (Å) | 4.068 (1) | 4.052 (9) | 4.51 | 4.564 | 4.603 | 6.75246 (10) | 3.9308 (3) | 3.9469 (8) |
| $c$ (Å) | 11.794 (3) | 12.787 (8) | 9.81 | 10.02 | 10.565 | 10.47440 (14) | 12.574 (2) | 12.559 (5) |
| $T_S$ (K) | - | - | 470 | ≃275 | 270 | - | - | - |
| $T_c$ (K) | 2.9 | 2.45 | 5.2 | 1.67 [1]/1.33 [2] | 2.1 | 1.8 | 1.0 (0.04) | 2.1 (0.04) |
| $H_{c2}$ (T) | 1.4 | ~0.2 | 2.5 | 0.85 [1,ab]/0.58 [2,ab] | 0.1 | 0.27 | 0.037 | 0.041 |
| | - | - | - | 0.52 [1,c]/0.33 [2,c] | - | - | - | - |
| $\gamma_0$ (mJ/(mol·K$^2$)) | 7.03 | 14.6 | 9.72 | 7.42 [1] | 9.2 | 8.6 (2) | 9.2 (0.3) | 9.3 (0.6) |
| $\Delta C/\gamma T_c$ | 0.91 | 1.36 | 1.67 | 1.27 [1] | 1.29 | 1.37 | 1.17 | 1.41 |
| References | [15] | [25] | [13] | [16] | [18] | [19] | [12] | [12] |

| Compound | $LaPd_2Sb_2$ | $LiCu_2P_2$ | $LaRu_2As_2$ | $LaRu_2P_2$ | $SrPd_2Sb_2$ [#] |
|---|---|---|---|---|---|
| space group | P4/nmm | I4/mmm | I4/mmm | I4/mmm | P4/nmm [LT]/I4/mmm [HT] |
| $a$ (Å) | 4.568 (4) | 3.8888 | 4.182 (6) | 4.031 (1) | 4.637[LT]/4.621[HT] |
| $b$ (Å) | 4.568 (4) | 3.8888 | 4.182 (6) | 4.031 (1) | 4.637[LT]/4.621[HT] |
| $c$ (Å) | 10.266 (2) | 9.5620 | 10.590 (3) | 10.675 (5) | 10.629[LT]/10.776[HT] |
| $T_c$ (K) | ≤1.4 | 4.1 | 7.8 | 4.1 | 1.9[LT]/0.6[HT] |
| $H_{c2}$ (T) | 0.58 | - | 1.6 | 0.114 [c/ab] | 0.6[LT]/0.06[HT] |
| $\gamma_0$ (mJ/(mol·K$^2$)) | 6.89 | - | - | - | 8.36[LT]/5.37[HT] |
| $\Delta C/\gamma T_c$ | 1.325 | - | - | - | ~1.4[LT]/1.43[HT] |
| References | [21] | [14] | [23] | [6,39] | [22] |

[ab] $H//ab$. [c] $H//c$. * Two superconducting transitions in $BaPt_2As_2$. [1] The first superconducting transition at a higher temperature. [2] The second superconducting transition. [#] Two superconducting phases in $SrPd_2Sb_2$ with different structures. [LT] Superconducting phase with the $CaBe_2Ge_2$-type structure. [HT] Superconducting phase with the $ThCr_2Si_2$-type structure.

### 2.3.1. $LaRu_2P_2$

$LaRu_2P_2$ was found to be superconducting at about 4 K almost 30 years ago [6]. The analysis of normal and superconducting states of $LaRu_2P_2$ single crystal demonstrates that $LaRu_2P_2$ is a rather conventional type II superconductor [39]. The isotropic SC also means the 3D nature of its FS topology. This FS property is later proved by the ARPES measurements performed on $LaRu_2P_2$ single crystals [47]. All bands that cross fermi energy ($E_F$) disperse strongly in the $k_z$ direction, hence their FS sheets are highly 3D. An excellent agreement can be achieved between ARPES spectra and calculated electronic structure using density functional theory (DFT). Thus, the resulted bandwidth renormalization is almost negligible, in contrast to that in iron-based superconductors, whose bandwidths are renormalized by a factor of 2–5 [48,49]. This also means that the electron correlation in $LaRu_2P_2$ is weaker. Figure 3 shows the FS topology of $LaRu_2P_2$. One can see that it is significantly different from that of its counterpart in superconducting iron pnictides.

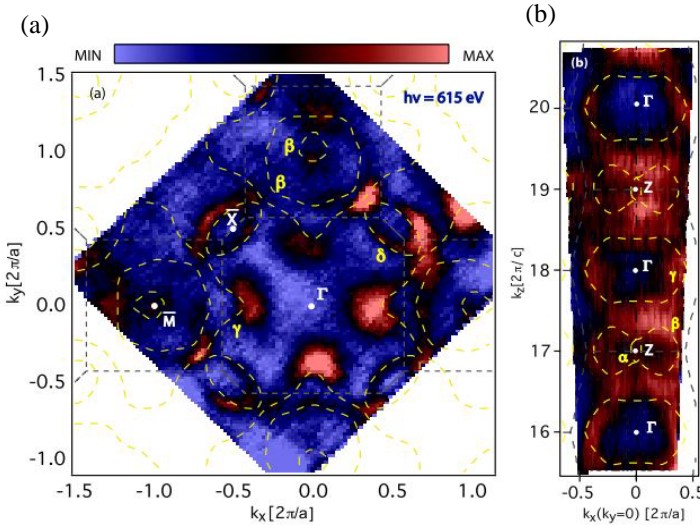

**Figure 3.** FS intensity maps obtained with soft X-ray ARPES for LaRu$_2$P$_2$. (**a**) and (**b**) are FS maps near the ($k_x$, $k_z$, 22 × 2$\pi$/c) plane and in the ($k_x$, 0, $k_z$) plane respectively. The superimposed dashed lines are the FS from density functional theory (DFT) calculation. $\bar{M}(\bar{X})$ is close to, but slightly different from, $M(X)$ in $k_z$ value. Pictures are from Reference [47].

It is interesting to mention that the bond distance between interlayer phosphor ions $d_{p-p}$ is believed to be particularly important, which identifies whether SC will occur in the 122 phosphides [12]. Theoretically, the critical distance for bond formation is about $d_{p-p} \sim 2.8$ Å between the interlayer P atoms [50]. An empirical rule is that a covalent P–P bond between adjacent planes is detrimental to the appearance of SC. Experimentally, SC were discovered in phosphides such as BaNi$_2$P$_2$ etc., and all these compounds were characterized by the absence of P–P bonds [6,9,12,44]. Moreover, alternating current susceptibility studies under hydrostatic pressure on single crystalline LaRu$_2$P$_2$ reveal a sudden disappearance of SC above 2.1 GPa, which is likely due to the formation of a covalent P-P bond between neighboring Ru$_2$P$_2$ planes [51]. However, exceptions like SrNi$_2$P$_2$ also exist, which shows SC in the collapsed tetragonal phase under pressure where there is a bond formation between the layers [8]. Thus the relationship between interlayer bonding and the mechanism for SC becomes controversial and needs more theoretical and experimental investigations.

### 2.3.2. $A$Ni$_2$$Pn_2$

BaNi$_2$As$_2$ and SrNi$_2$P$_2$ undergo a structural phase transition at 130 K and 325 K, respectively, and bulk SC was observed at $T_c = 0.68$ K and 1.4 K [8]. As shown in Figure 4a, the heat capacity data of these two compounds can be well fitted by a BCS *s*-wave expression [40,42,52]. Moreover, the concave field responses of residual linear term $\kappa_0/T$, determined from the thermal conductivity measurements, give clear evidence that both BaNi$_2$As$_2$ and SrNi$_2$P$_2$ are fully gapped single gap superconductors [52]. Theories from first principle calculations also give consistent results for BaNi$_2$As$_2$, which shows a FS distinct from BaFe$_2$As$_2$ [53]. Through partial substitution of P for As in BaNi$_2$As$_2$, the triclinic phase transition is strongly suppressed, accompanying a stepwise enhancement of $T_c$ from 0.6 K to 3.3 K [54]. By detailed investigation of specific heat results, the authors argue that the chemical pressure effect induced by phosphorous doping results in the softening of phonon, which could enhance the electron–phonon coupling, and thus SC. A recent ARPES study (Figure 5b,c) exhibits a good agreement with the theoretical calculations (Figure 5a), which indicates a rather weak electronic correlation in SrNi$_2$As$_2$ compared to that in iron-based pnictides [55]. The FS topology of $k_z \sim 0$ plane (Figure 6b) shows very similar behavior with that of $k_z \sim \pi$ plane (Figure 5c), except for an additional hole pocket observed around $Z$ point. This means a strong 3D character of FS in SrNi$_2$As$_2$, which is further proved by the transport property measurements for BaNi$_2$P$_2$ [41]. The apparent different

size between electronic and hole pockets (for instance, the small hole pocket at $\Gamma$ point and the nearly hexagonal electron pocket around $M$ point in Figure 5b) infers the absence of good Fermi nesting conditions; thus, it is difficult to form a SDW in this compound, which strongly contrasts the results obtained in iron-based superconductors. This issue is also discussed from theory [56,57]. As is well known, the FS of the iron-based supercondutors is formed by cylindrical hole sheets near the $\Gamma$ point and electron sheets at the corners of the Brillouin zone (see Figure 5d–f) [58]. This kind of electronic structure makes the system unstable and tends to form charge/spin ordering, which is often caused by structure distortions. In addition, compared with the FS of $SrNi_2As_2$, the FS sheets in iron-based superconductors is more like quasi-2D in shape.

Compared to the quasi-2D FSs of iron-based superconductors, the FSs of nickel-based superconductors are more complicated and 3D in nature, which may account for the differences between these two systems [43,53,59–61]. Angle-resolved photoemission spectroscopy (ARPES) study performed on the end member-$BaNi_2P_2$ shows that both hole and electron FS pockets are observed, and the shapes of the hole FSs dramatically change with photon energy, which means a strong three-dimensionality [62]. This observation is also consistent with band-structure calculations and de Haas–van Alphen (dHvA) measurements, which further confirms the 3D nature of the FSs in $BaNi_2P_2$ [59]. Note that dHvA studies on $BaIr_2P_2$ give similar results of FS topology [63]. By comparing the parameters for different Ni-containing 122-IFPSs, as shown in Figure 6, Ideta et al. also propose that not only the 3D FSs but also the interlayer Pn-Pn bond distance ($z_{Pn-Pn}$) or the pnictogen height ($h_{Pn}$) may be important parameters to determine $T_c$ in the $ANi_2Pn_2$ family [62].

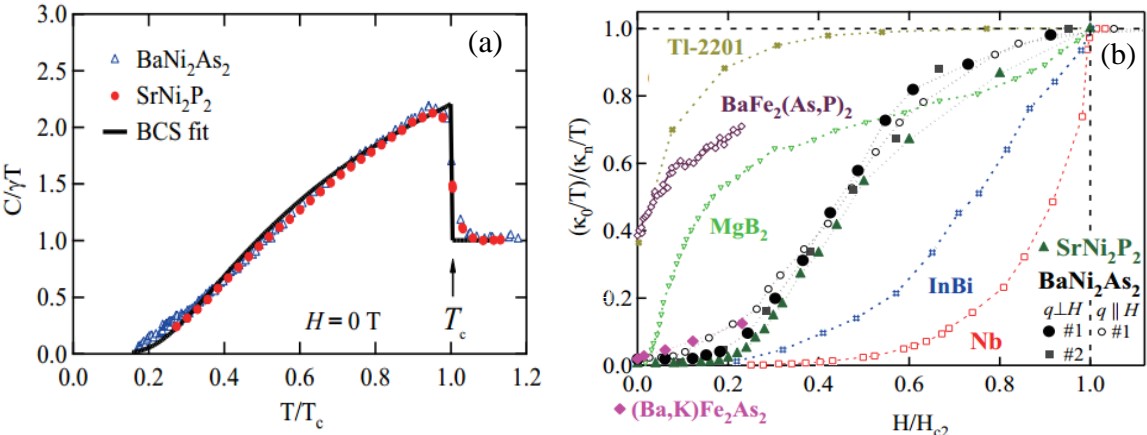

**Figure 4.** (a) Temperature dependence of scaled $C/T$ for $BaNi_2As_2$ and $SrNi_2P_2$ under zero field, with the solid curve being a Bardeen–Cooper–Schrieffer (BCS) fit. (b) Scaled residual linear term of $(\kappa_0/T)/(\kappa_n/T)$ versus $H/H_{c2}$ of $BaNi_2As_2$ for sample 1 (# 1, $H \perp I$ and $H//I$) and # 2 with $H \perp I$, and $SrNi_2P_2$ ($H \perp I$). Data for Nb (clean, fully gapped *s*-wave), InBi (dirty, fully gapped *s*-wave), $MgB_2$ (multi-band gap), and Tl-2201 (*d*-wave with line nodes) are shown for comparison. Pictures are from Reference [52].

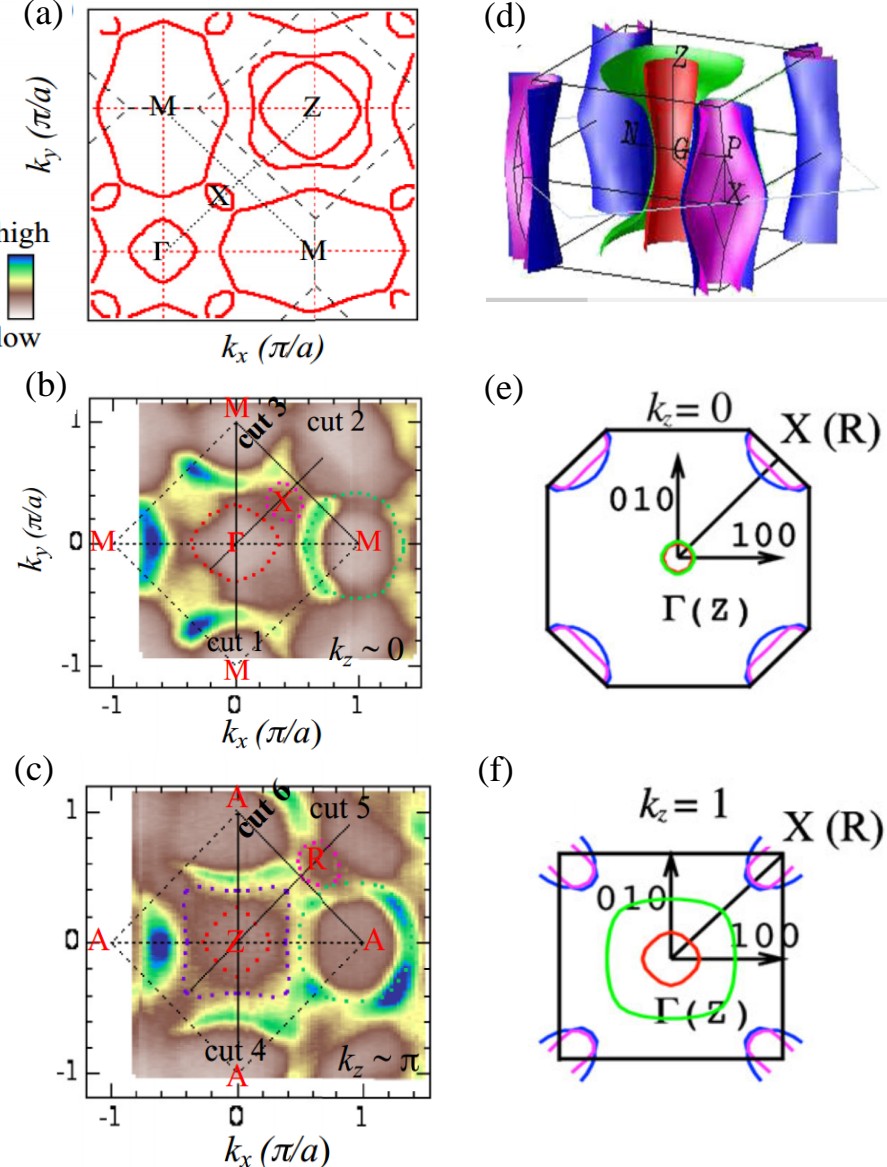

**Figure 5.** (**a**) FS of SrNi$_2$As$_2$ single crystal obtained by local density approximation calculations. (**b**) and (**c**) are ARPES mapping intensity plots at Fermi energy recorded at 30 K in the $k_z \sim 0$ and $k_z \sim \pi$ planes, respectively. (**d**) 3D FS of BaFe$_2$As$_2$ obtained from full-potential linearized plane wave (FLAPW) calculations. (**e**) and (**f**) are calculated FS cross-sections for $k_z = 0$ ($X - \Gamma$ plane) and $k_z = 1$ ($R - Z$ plane) for BaFe$_2$As$_2$ using the same method same as (d). Note that the FS derived from ARPES measurements for BaFe$_2$As$_2$ is generally in agreement with the FLAPW calculations. Pictures are from Reference [55,58].

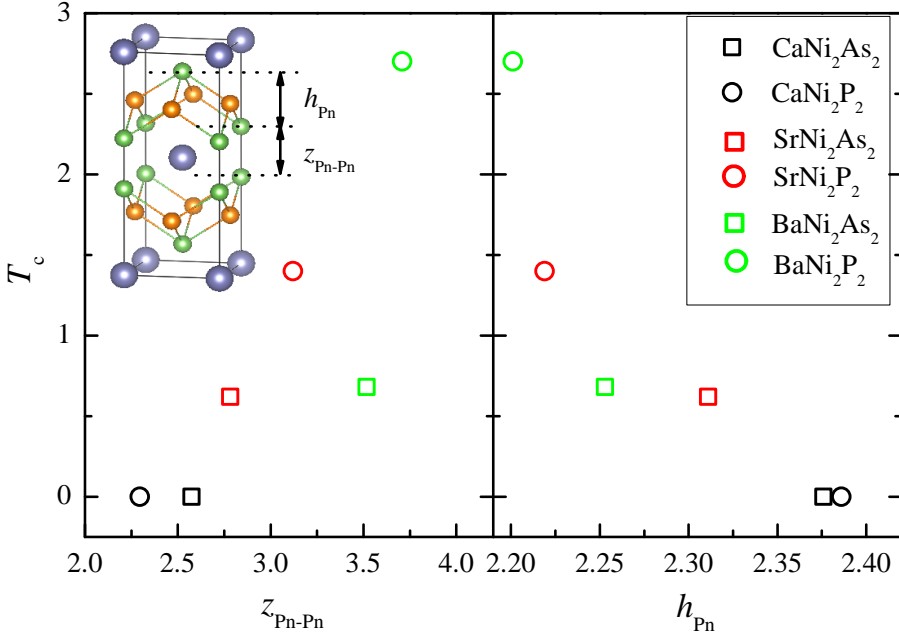

**Figure 6.** Relationship between $T_c$, $z_{Pn-Pn}$, and $h_{Pn}$ for $A$Ni$_2$Pn$_2$ compounds, where $A$ = Ba, Sr, Ca; $Pn$ = P, As. The data values for $z_{Pn-Pn}$ and $h_{Pn}$ are taken from Reference [64].

### 2.3.3. $A$Pt$_2$Pn$_2$

All the $A$Pt$_2$Pn$_2$ ($A$ = Sr, Ba; $Pn$ = As, Sb) superconductors possess CaBe$_2$Ge$_2$-type structure or a crystal structure related to it. Measurements of physical properties were first reported for SrPt$_2$As$_2$, which show a CDW transition at high temperature while it enters into superconducting state below 5.2 K [13]. Detailed heat capacity study, combined with nuclear magnetic resonance (NMR) measurement, suggests a Fermi liquid state in SrPt$_2$As$_2$ [65]. As shown in Figure 7a,b, by fitting the specific heat jump ($\Delta C$) using different gap functions, one can see that the two s-wave gap functions best capture the temperature evolution of $\Delta C$ in the whole temperature range below $T_c$. This conclusion is further evidenced by the dependence of Sommerfeld coefficient ($\gamma$) as magnetic fields ($H$) (see Figure 7c). It is known that in a type II s-wave superconductor, vortex core mainly contribute the thermal excitations and the number of the vortices grows linearly in $H$; thus, $\gamma$ increases linearly with $H$. However, in SrPt$_2$As$_2$, $\gamma(H)$ can be decomposed into two linear terms, which implies a two-gap model reminiscent of a prototypical two-gap superconductor MgB$_2$ [66]. Therefore, SrPt$_2$As$_2$ is argued to be another representative of two-gap superconductors [65]. In addition, the spin-lattice relaxation rate $1/T_1$ exhibits a linear temperature dependence (see Figure 7d), which is expected for Fermi liquid. Together with the $T^2$ resistivity at low temperatures, SrPt$_2$As$_2$ manifests itself as a BCS superconductor.

BaPt$_2$As$_2$ exhibits two subsequent superconducting transitions at $T_{c1}$ ~ 1.67 K and $T_{c2}$ ~ 1.33 K [16]. These two transitions are found to be from the same origin. Moreover, low-temperature specific heat analysis supports a nodeless BCS-type SC for BaPt$_2$As$_2$. Similar BCS-like SC with moderate coupling is also proposed in SrPt$_2$Sb$_2$ and BaPt$_2$Sb$_2$ [18,19].

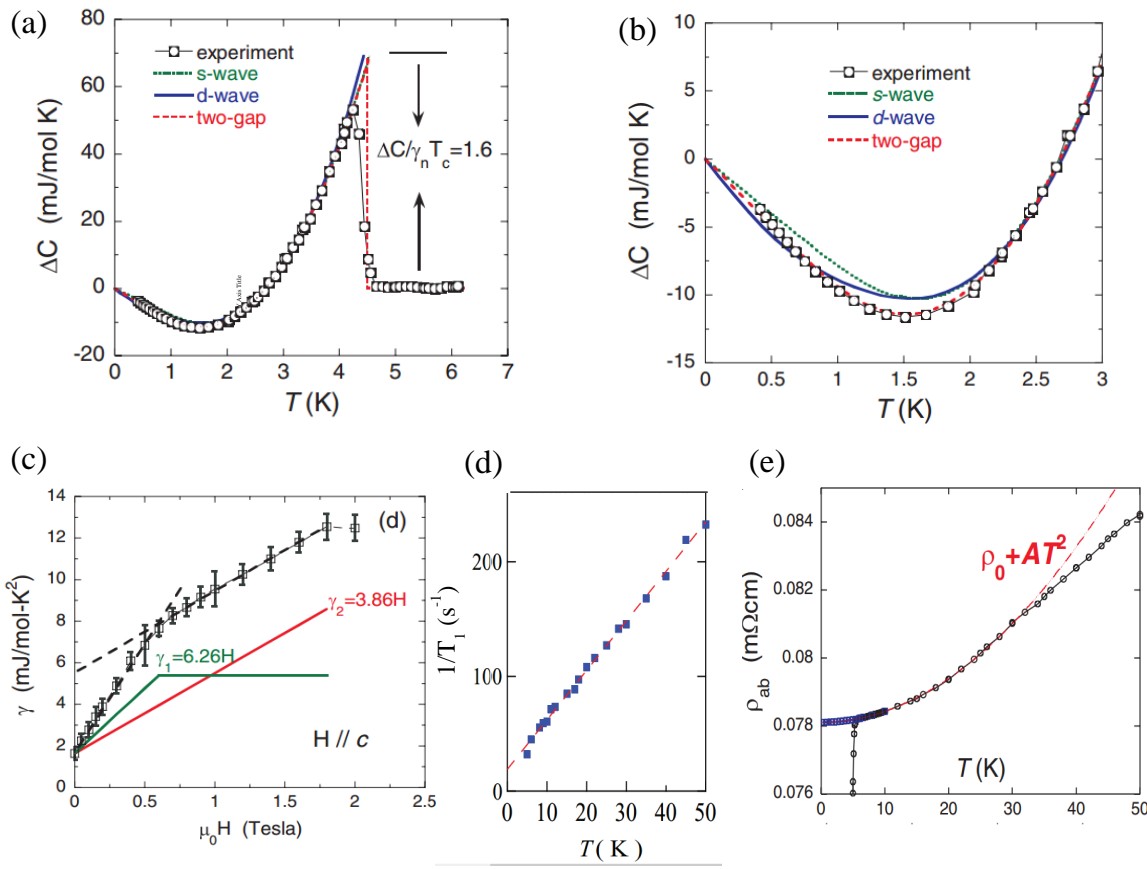

**Figure 7.** Physical properties of $SrPt_2As_2$. (**a**) is $\Delta C$ as a function of $T$, plotted with three fitted results from different gap functions. As can be seen, the two-gap model fits the experimental data best. (**b**) is an expanded view of (a) below $T_c$. (**c**) The plot of $\gamma$ versus $H$. The slope of the changing rate of $\gamma$ with increasing $H$ can be described well by two different constants. (**d**) Temperature dependence of $^{195}$Pt nuclear spin-lattice relaxation rate $1/T_1$. The dashed line indicates a linear fit. (**e**) An enlarged plot of low-temperature resistivity, with blue squares denoting the value extracted from linearly extrapolating $\rho(T, H)$ above $H_{c2}$ to $H = 0$. The red dashed line is a fit to $\rho(0) + AT^2$. Pictures are from Reference [65].

### 2.3.4. $APd_2Pn_2$

All $APd_2As_2$ compounds with $A$ being Ca, Sr and Ba superconduct at low temperatures, whilse $BaPd_2As_2$ has the highest $T_c$ at 3.85 K among them [17,24]. Despite the difference in crystal structure between $BaPd_2As_2$ and $(Sr/Ca)Pd_2As_2$, these three compounds show similar physical properties both in the normal and superconducting states, and the parameters defined from the measurements have small discrepancy (see Table 1). Furthermore, all three of them are believed to be conventional $s$-wave superconductors from transport property analysis [17,24,67]. Extra solid proof comes from the NMR measurements for $CaPd_2As_2$ [67]. The spin-lattice relaxation rate $1/T_1$ varies proportionately to temperature, following the Korringa relation $1/T_1T = $ constant, resembling the behavior observed in the aforementioned $SrPt_2As_2$ [65]. Besides, through detailed analysis on the $1/T_1$ and Knight shift $K$, Ding et al. came to the conclusion that spin correlations are absent in this compound [67]. This is in contrast to iron-based superconductors, in which AFM fluctuations are believed to be important for the occurrence of unconventional SC. In the superconducting state, a clear coherence peak is observed in the $1/T_1 - T$ curve. Below the temperature when the peak shows up, $1/T_1$ decreases exponentially as lowering temperatures, indicative of BCS SC [67].

$SrPd_2Sb_2$ has two different kinds of polymorphs: a low-temperature phase with a tetragonal $CaBe_2Ge_2$-type structure and a high-temperature phase with $ThCr_2Si_2$-type structure [22]. These two

phases enter into the superconducting state at $T_{c1} = 1.95$ K and $T_{c2} = 0.6$ K, respectively. The specific heat data below $T_c$ can be well fitted by a conventional BCS function, which suggests a weak electron–phonon coupling in $SrPd_2Sb_2$.

## 3. Relationship between Structure and $T_c$

It is an empirical strategy that we may find new superconductors with higher $T_c$ more precisely if the relationship between structure property and $T_c$ can be found in the system. In iron-based superconductors, structure parameter dependence of $T_c$ has been proposed in several systems [68,69]. The As-Fe-As bond angle $\alpha$ was revealed to strongly correlate with superconducting property, and maximum $T_c$ value seems to selectively appear around $\alpha = 109.47°$; i.e., when $FeAs_4$-tetrahedrons form a regular shape [68]. This indicates a clear relationship between crystal structure and SC. Anion height (from Fe-layer) dependence of $T_c$ for typical iron-based superconductors has been reported by Mizuguchi et al. in Reference [69], while a symmetric curve with a peak near 1.38 is suggested to be obeyed by these superconductors both under ambient and high pressures. In terms of this, we did a detailed analysis of the possible relation between crystal structure and SC in 122-IFPSs. However, we could not identify similar dependency of SC on structure parameters as obtained in iron pnictides superconductors. The aforementioned enhancement of $T_c$ with the increase of Pn-Pn bond distance in $ANi_2Pn_2$ family is only limited in Ni-containing 122-IFPSs. As 122-IFPSs exhibit various different crystal structures, this may be the reason that it is difficult to find a uniform relation between the structural parameters and SC.

## 4. Summary and Outlook

Through the above experimental review for 122-IFPSs, one can easily come to the conclusion that the 122-IFPSs are obviously different from their iron-based pnictides counterparts in SC mechanism. On one hand, magnetism is absent in 122-IFPSs, while in iron-based pnictides, AFM spin correlation is believed to be of significant importance for unconventional SC. On the other hand, all the measurement results point to a conventional BCS phonon-mediated SC in 122-IFPSs. However, a sign-reversing $s\pm$ gap structure and multiple energy gaps are proposed for iron-based superconductors, albeit the underlying pairing mechanism still remains controversial. In spite of this, the exploration and investigation of 122-IFPSs are particularly meaningful, not only for searching new superconductors in this field, but also for getting a deeper understanding of the mechanism for SC in iron-based pnictides. Besides, by detailed comparison between these two systems, both from similarities and differences, one may find a rational route to some interesting materials.

In conclusion, we have briefly reviewed the experimental results of 122-IFPSs. BCS-type SC with weak or moderate coupling are proposed for 122-IFPSs. Strong evidence comes from resistivity, magnetic susceptibility, and specific heat measurements. To be specific, fitting for the specific heat data below $T_c$ for $BaNi_2As_2$, $SrPt_2As_2$ and $CaPd_2As_2$, etc. together with the parameters obtained (such as Sommefeld coefficient and the value of $\Delta C / \gamma T_c$), suggests a conventional $s$-wave pairing states in these compounds [17,22,52,65]. More direct proofs are derived from ARPES measurements for $LaRu_2P_2$, $(Sr/Ba)Ni_2As_2$ and $BaNi_2P_2$; NMR measurements for $CaPd_2As_2$ and $SrPt_2As_2$; field-dependent thermal conductivity measurements for $BaNi_2As_2$ and $SrNi_2P_2$; and de Haas–van Alphen measurements for $BaNi_2P_2$ and $BaIr_2P_2$ [34,47,52,55,59,62,63,65,67]. However, the relationship between 122-IFPSs and iron-based pnictides, especially the similarities of structure property and the variation of $T_c$ with doping or pressure effects, still needs more experimental and theoretical work to clarify.

**Acknowledgments:** This work was supported by the Fundamental Research Funds for the Universities of China.

**Author Contributions:** Pan Zhang analyzed the data and wrote the paper; Hui-fei Zhai helped collecting related data and did the final check on the manuscript. Both authors have read and approved the final manuscript.

**Conflicts of Interest:** The founding sponsors had no role in the design of the study; in the collection, analyses, or interpretation of data; in the writing of the manuscript, and in the decision to publish the results. Both authors declare no conflict of interest.

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
