# Peer review of "Superconductivity in 122-Type Pnictides without Iron"

_condensedmatter, doi:10.3390/condmat2030028_

Reviewer 1 Report

Pan Zhang and Hui-Fei Zhai review the properties of the 122 iron-free pnictides by collecting and comparing the current (experimental) available results. The results are described properly and the significance of the content is very high. The authors successfully included the necessary literature for a comprehensive review of the specific subject. 

However, an extensive editing of the English language is needed since many grammatical errors were detected.

Hence, I recommend the acceptance of the manuscript after the language corrections.

Author Response

Thank the referee for the positive comments. Our detailed reply can be found in the attached pdf file. Thank you for your time. 

Reviewer 2 Report

This review article focuses on the Fe-free 122-type pnictide superconductors. After the discovery of FePn-122 superconductors, many superconductors with transition metals other than Fe have been discovered. Considering the structural similarities to the FePn-122 superconductors, we simply expect the emergence of high Tc superconductivity. Therefore, summarizing the reports on the Fe-free 122 superconductors and discussing the differences in superconductivity between FePn-122 and Fe-free 122 superconductors are very important work. Since this paper contains reviews on crystal structure, Tc, Hc2, structural transition, CDW states, and electronic states, I consider that the readers of Condensed Matter will get some fruitful insights from this article; hence, I recommend publication of this article after minor revisions. Please address the points listed below.

1)    On the introduction part, please separate it into two paragraphs. New line from “On the other hand, SC is…” may be suitable.

2)    On Fig. 1, adding bonding between T and Pn may be useful for readers to understand the structural characteristics.

3)    On the third section, “Relationship between structure and Tc”, the authors just mentioned that there is no correlation between Tc and local structure. On the other hand, in the Ni-based system, there is correlation. Can the authors see some correlation, if looking the same transition metal series only.

4)    There are still errors of English. Please check it again in whole.

5)    In the affiliation, e-mail address different from that of corresponding author appears. Please delete.

Author Response

(The authors gave the same response as above.)

Reviewer 3 Report

In this paper, the authors review the electronic properties of the iron-free 122-type superconductors. The crystal structure, structural phase transition, superconducting properties in many materials in this system are summarized. Especially, the superconducting states are discussed in Ni-, Pt- and Pd-based 122 type pnictide materials. The authors conclude that the conventional superconductivity originating from BCS mechanism appears in these materials, which is in contrast to that in the iron-based superconductors.

I have the following comments regarding the present manuscript.

(1) In Fig. 5(b), the result of MgB2 is not shown. Please add the data of MgB2 in this figure.

(2) In Fig. 8(e), the label of the y-axis should be fixed.

Based on the above, I think that the paper provides interesting information to the community working in this field, but the manuscript is not suitable for the publication in Condensed matter until the figures are revised.

Author Response

(The authors gave the same response as above.)
